# The validation of the CTS-6 Evaluation Tool for diagnosing carpal tunnel syndrome (CTS) in Thai wheelchair users

**Montana Buntragulpoontawee**[1,2], **Siam Tongprasert**[3], **Jiraporn Khorana**[2,4,5], **Kittipong Kitisak**[1,2], **Waris Karinuntakul**[3], **Sineenard Pornjaksawan**[1], **Phichayut Phinyo**[2,6]*

1 Neuro-Mobility Unit, Department of Rehabilitation Medicine, Faculty of Medicine, Chiang Mai University, Chiang Mai, Thailand, 2 Center of Clinical Epidemiology and Clinical Statistics, Faculty of Medicine, Chiang Mai University, Chiang Mai, Thailand, 3 Department of Rehabilitation Medicine, Faculty of Medicine, Chiang Mai University, Chiang Mai, Thailand, 4 Division of Pediatric Surgery, Department of Surgery, Faculty of Medicine, Chiang Mai University, Chiang Mai, Thailand, 5 Clinical Surgical Research Center, Faculty of Medicine, Chiang Mai University, Chiang Mai, Thailand, 6 Department of Biomedical Informatics and Clinical Epidemiology (BioCE), Faculty of Medicine, Chiang Mai University, Chiang Mai, Thailand

* phichayut_phinyo@cmu.ac.th

## Abstract

### Objective

The CTS-6 Evaluation Tool is a clinical diagnostic tool for carpal tunnel syndrome. It was originally developed using data from non-disabled individuals and has never been validated in different population. This study aimed to validate CTS-6's diagnostic performance at a cutoff score of 12 in a new population of wheelchair users.

### Methods

The participants were 54 Thai wheelchair users from a university hospital's neuropathy registry. Those with a history of nerve injury, fracture, neuropathy, or pregnancy were excluded from the study. All underwent clinical exam, CTS-6, and electrodiagnosis (blinded evaluator). Carpal tunnel syndrome was diagnosed based on clinical symptoms and electrodiagnostic criteria.

### Results

Of 54 participants, 13 were female (24.1%) with an average age of 46.9 (SD 12.2) years, and 18 (33.3%) participants had carpal tunnel syndrome. Duration of disability (years) was significantly longer in cases; median 24 ($Q_1$ 19.0, $Q_3$ 28.9), $p < 0.001$. Discriminative performance: Area under the receiver operating curve 0.935 (95%CI:0.891-0.978). At the 12 cutoff point, the sensitivity was 43.8% (95%CI:26.4-62.3%) and the specificity was 100.0% (95%CI:94.8-100.0%). A lower cutoff point showed increased sensitivity and specificity. Symptomatic subgroup analysis showed similar diagnostic performances.

**Data availability statement:** All relevant data are within the paper and its Supporting Information files.

**Funding:** The Faculty of Medicine at Chiang Mai University provided a grant for The Entrapment Neuropathy in Thai Wheelchair Users due to SCI Registry. (grant number 046/2565). The funders had no role in study design, data collection and analysis, decision to publish, or preparation of the manuscript.

**Competing interests:** The authors have declared that no competing interests exist.

## Conclusion

The CTS-6 Evaluation Tool is a simple clinical diagnostic tool that does not require sophisticated investigation. The CTS-6's discriminative ability remains strong. The diagnostic performance at a cutoff score of 12 showed moderate sensitivity and high specificity. Applying a cutoff score of 12 could help rule in the diagnosis where access to electrodiagnosis is limited. A lower cutoff score that is 7.5 could be applied as a screening test to rule out the diagnosis, as it provides moderately higher sensitivity at the cost of increased false positives.

## Introduction

Carpal tunnel syndrome, or CTS, is the most common entrapment neuropathy in the upper extremities, resulting from median nerve compression at the wrist, which results in repetitive or forceful wrist movement causing median nerve compression at the wrist [1,2]. Symptoms of CTS include pain, a tingling sensation, numbness, or, in severe cases, weakness and loss of function in the affected hand [3,4].

Individuals with limited lower-extremity functional or structural control, such as paraplegia following spinal cord injury, require wheelchair use as their principal method for household and community mobility. As wheelchair users rely heavily on their upper extremities for mobility and self-care activities, this population is at risk for developing CTS [5,6]. Many previous publications also showed a higher prevalence of median neuropathy at the wrist or CTS in wheelchair users (22% to 64.6%) than in non-disabled persons (approximately 3% in females and 2% in men) [2,7–11]. For making a diagnosis of entrapment neuropathy, history taking, and physical examination are the first step [12,13]. Sometimes, a confirmatory electrodiagnostic exam is requested. Considering the heavy reliance of wheelchair users on upper extremity function, a timely and accurate clinical diagnosis of CTS would assist in guiding care and proper electrodiagnostic referral. Hence, a systematic clinical evaluation tool called the CTS-6 was developed. The tool aims to guide clinicians in making a CTS diagnosis regardless of their expertise level [14,15]. The tool consists of structured history taking and physical examination with a specified score for each item; a total score > 12 indicates an 80% probability of having CTS [14,15].

The previous CTS-6 validation studies by Fowler et al. [16], Makanji et al. [17], and Wang et al. [18] was conducted in non-disabled individuals, evaluating different cutoffs at 12, 14,16 and 18. These showed varying diagnostic performance results, with sensitivity ranging from 60 to 90%, and specificity ranging from 59 to 91%. However, to the authors' knowledge, no prior CTS-6 diagnostic performance validation has been performed in wheelchair users, the population domain with higher CTS prevalence.

Therefore, this study aimed to validate the performance of the CTS-6 Evaluation Tool at a cutoff score of 12 for diagnosing CTS in a new population of wheelchair users and to investigate different cutoff points for varying diagnostic performances, leading to potentially different roles in its clinical application.

## Materials and methods

### Patient domain

The study retrieved data from The Entrapment Neuropathy in Thai Wheelchair Users due to SCI Registry. The registry participants were Thai wheelchair users who visited Maharaj Nakorn Chiang Mai Hospital, a tertiary hospital in a medical school setting, for clinical service

and willingly participated in the registry from 31 August 2021 to 17 May 2023 [19]. All wheelchair users must be at least 18 years of age and have been using wheelchairs as a principal method of daily mobility for at least three months before participating in the registry. Participants with known peripheral neuropathy and systemic diseases predisposing to neuropathy, such as diabetes mellitus, thyroid disease, or pregnancy in females, were excluded. Other local events potentially predispose the median nerve of the wrist to injury; for example, previous fractures or joint deformities were excluded.

Written informed consent was obtained from all the participants. The rehabilitation specialist performed clinical evaluations, including history-taking and physical examination for CTS.

The registry and this study were approved by the research ethics committee, Faculty of Medicine, Chiang Mai University; Study Code No.: REH-2564-08339/Research ID: 8339 and Study Code: REH-2565-08965/Research ID: 8965. The Faculty of Medicine at Chiang Mai University provided a grant for the registry (grant number 046/2565). The authors declare that they have no conflicts of interest. This study adhered to the STARD 2015 guidelines for reporting diagnostic accuracy studies [20,21].

## Measures

**Spinal Cord Independence Measure.** The Spinal Cord Independence Measure (SCIM) is a quantitative evaluation tool for describing patients with SCI's ability to perform activities of daily living (self-care sub-score), move around between locations necessary for basic daily activities (mobility sub-score), as well as breathing and bowel/bladder function (respiratory and sphincter). The self-care sub-score ranges from to 0-20, the mobility sub-score ranges from to 0-40, the respiratory and sphincter sub-scores ranges from to 0-40, and the total score is 100. The higher the score, the greater the ability [22].

**Boston Carpal Tunnel Syndrome Questionnaire.** The Boston Carpal Tunnel Syndrome Questionnaire (BCTQ) describes the severity of the symptoms and functional disturbances resulting from CTS. The symptom severity score (BCTQ-SSS) ranges from 11 to 55, with higher scores representing more severe symptoms. The functional severity score (BCTQ-FSS) ranges from 8 to 40, with higher scores representing more significant functional deficits [23].

**CTS-6 EvaluationTool.** The CTS-6 Evaluation Tool was structured to assist clinicians in diagnosing CTS. It was developed in 2006 by Graham et al. [14], a group of hand surgeon experts. The tool consisted of two symptoms and history items and four physical examination items; each positive item was scored differently, with a score of > 5 equaling 0.25 probability of carpal tunnel syndrome and a score of > 12 equaling 0.80 probability of carpal tunnel syndrome. The items and scores of the CTS-6 Evaluation Tool are listed in Table 1.

**Electrodiagnosis study.** Rehabilitation physicians performed an electrodiagnostic study using the machine brand Dantec by Medtronic (Keypoint Workstation model number 22022). A hand-held infrared thermometer measured skin temperature at the hands and kept it above 32 °C with a hot pack when necessary before performing the electrodiagnostic study.

Electrodiagnostic study techniques, including antidromic median and ulnar sensory and motor nerve conduction study, Combined Sensory Index, and median-ulnar lumbrical-interossei motor nerve conduction study were performed. Electrodiagnostic criteria for CTS were applied according to the American Association of Neuromuscular and Electrodiagnostic Medicine publications [24,25].

All participants underwent the CTS-6 Evaluation Tool (index test) and electrodiagnostic evaluation to confirm CTS by separate examiners without prior knowledge of either test result. CTS diagnosis was confirmed (reference standard) when the electrodiagnostic findings of the symptomatic wrist corresponded to the electrodiagnostic criteria for CTS.

**Table 1. Items and scoring of the CTS-6 Evaluation Tool [14].**

| Symptoms and history | | Score |
|---|---|---|
| 1. | **Numbness predominantly or exclusively in the median nerve territory** <br> Sensory symptoms are mostly in the thumb, index, middle and/or ring fingers | 3.5 |
| 2. | **Nocturnal numbness** <br> Symptoms are predominantly the patient sleep; numbness wakes patient from sleep | 4 |
| **Physical Examination** | | |
| 3. | **Thenar atrophy and/or weakness** <br> The bulk the thenar area is reduced or where manual motor testing shows strength of grade 4 less | 5 |
| 4. | **Positive Phalen's test** <br> Flexion of the wrist reproduces her worsened symptoms of numbness in the median nerve territory | 5 |
| 5. | **Loss of 2-point discrimination** <br> Failure to discriminate 2 points held 5 mm or less apart from one another, in the median innervated digits | 4.5 |
| 6. | **Positive Tinel sign** <br> Light tapping over the median nerve at the level of the carpal tunnel causing radiating paraesthesias | 4 |
| | Total | (26) |

## Statistical analysis

Stata16 (StataCorp, College Station, TX, USA) was used for data analysis. The observation units for analysis were the person (for demographic data description) and wrist (for symptoms and CTS diagnosis). Categorical data were described as frequency percentage and tested for differences between groups using the chi-square test. Continuous data with normal distribution were described using the mean and standard deviation, and median and interquartile range were used for non-normally distributed data. The difference between the two groups was tested using an independent t-test for normally distributed data and the Mann-Whitney U test for non-normally distributed data. The discriminative ability of the CTS-6 Evaluation Tool for CTS diagnosis was assessed using the area under the receiver operating characteristic curve (AuROC). Sensitivity, specificity, positive predictive value (PPV), negative predictive value (NPV), and likelihood ratio (LR) were analysed to determine the diagnostic accuracy of the CTS-6 Evaluation Tool in the total wrist number and symptomatic/asymptomatic subgroup.

## Study size estimation

The sample size calculation for the CTS-6 study was based on pilot data. Two different cutoff points for diagnosing CTS were used; one was 12, as suggested in the original development article [15], and another was 10, as newly proposed for use in this study. There were 43 wrists in the pilot data, and 25 were wrists with CTS. The prevalence was 58.1%, and the sensitivity was 8% (cutoff point at 12) and 16% (cutoff point at 10), respectively. The specificity was 90% for both the cutoff points. The authors used a one-group proportion formula: d (acceptable margin of error) = 0.1. Calculation based on specificity, a sample size of 100 was required.

## Results

A total of 54 participants were included. A total of 101 wrists were included in the study; 7 were excluded, 6 because of previous fractures, and another because of nerve injury. All underwent clinical examination, an evaluation using the CTS-6 tool, and an electrodiagnostic study (Fig 1). There were 18 participants (33.3%), of whom 32 wrists (31.7%) had confirmed CTS. Most participants were male, 41 (75.9%), and 13 (24.1%) were females. The result did not show a female predisposition for having CTS. All participants were long-term wheelchair users with a median duration of disability of 18.0 years; the duration of disability

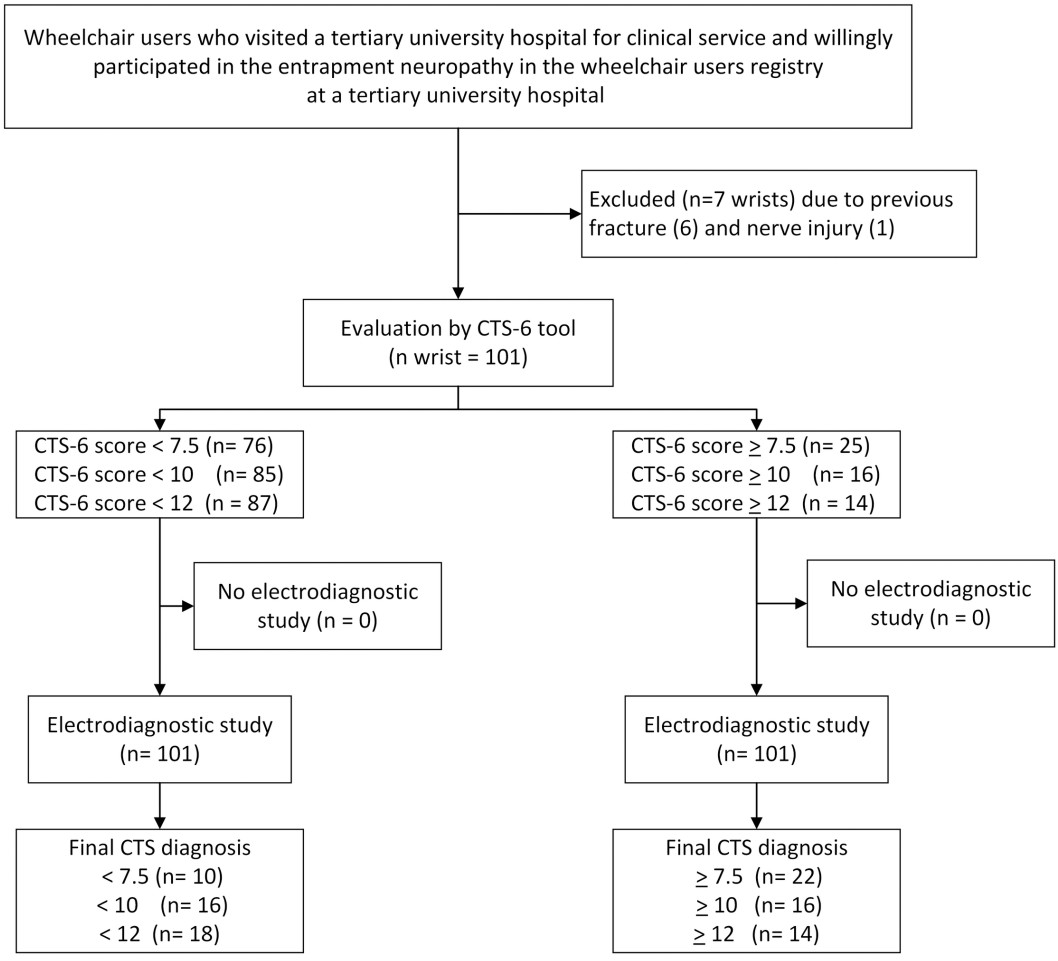

**Fig 1. Study flow diagram of wheelchair users evaluated for carpal tunnel syndrome (CTS).**

for participants with CTS (24.0 years) was significantly longer than for participants without CTS (12.9 years), p-value < 0.001. All the participants were independent of self-care activities. While higher BMI usually associates with CTS, mean body mass index seemed to be higher in participants with CTS ($23.9 \, kg/m^2$) compared to participants without CTS ($22.7 \, kg/m^2$); however, the difference did not reach statistical significance, p = 0.196 [26]. The CTS-6, BCTQ symptom severity and BCTQ functional severity scores were significantly higher in the participants with CTS (Table 2).

The overall discriminative ability of the CTS-6 Evaluation Tool for diagnosing CTS in wheelchair users was 0.935 (95%CI:0.891-0.978) (Fig 2). From the AuROC, the diagnostic performance of the CTS-6 Evaluation Tool at 12 cutoff score was validated and two lower cutoff points of 10 and 7.5 were examined. The original cutoff point at 12 showed the highest PPV of 100.0% (95%CI:76.8-100.0%) and specificity of 100.0% (95%CI:94.8-100.0%), giving no false negative results, while the sensitivity was 43.8% (95%CI:26.4-62.3%). The middle cutoff score of 10 showed less sensitivity at 50.0% (95%CI:31.9-68.1%), and the specificity remained high at 100.0% (95%CI:94.8-100.0%). The lowest cutoff of 7.5 showed a moderate sensitivity of 68.8% (95%CI:50.0-83.9%), and the specificity remained high at 95.7% (95%CI:87.8-99.1%) (Table 3). There were 64 asymptomatic wrists, and none of them were diagnosed with CTS. Of the remaining 37 symptomatic wrists, 32 were confirmed as having CTS. Sensitivity

**Table 2. Characteristics of wheelchair users evaluated for carpal tunnel syndrome (CTS).**

| Clinical characteristics (persons) | Total (54) | CTS (n = 18) | No CTS (n = 36) | P-value |
|---|---|---|---|---|
| Age (years)* | 46.9 (12.2) | 52.7 (9.9) | 44.1 (12.4) | 0.014 |
| Female sex, n (%) | 13 (24.1) | 6 (33.3) | 7(19.4) | 0.319 |
| Self-Care SCIM score (Q1 Q3) | 20 (20,20) | 20 (20,20) | 20 (20,20) | 0.634 |
| Mobility SCIM score* | 17.8 (1.2) | 18 (0.9) | 17.75 (1.3) | 0.467 |
| Respiratory and Sphincter SCIM score | 23 (19.0,30.0) | 23 (19.0,30.0) | 26 (20.0,30.0) | 0.794 |
| Total SCIM score | 61.5 (57.0,69.0) | 61.0 (57.0,69.0) | 62.5 (57.0, 68.0) | 0.868 |
| Duration of disability (years) | 18.0 (8.7,25.2) | 24.0 (19.0, 28.9) | 12.9 (3.8, 20.8) | <0.001 |
| BMI (kg/m$^2$)* | 22.7 (4.5) | 23.9 (5.8) | 22.2 (3.7) | 0.196 |
| Rt dominant hand, n(%) | 48 (88.9) | 14 (77.8) | 34 (94.4) | 0.087 |
| **Clinical Characteristics (Wrists)** | **Total** | **CTS (n = 32)** | **No CTS (n = 69)** | **P-value** |
| CTS-6 score | 0 (0,5) | 10.3 (4, 12.5) | 0 (0,0) | <0.001 |
| BCTQ symptom severity score | 11 (11,13) | 14 (11,11) | 11 (11,11) | <0.001 |
| BCTQ functional severity score | 8 (8,8) | 8 (8,8) | 8 (8,8) | 0.003 |

Data presented as median (1$^{st}$ to 3$^{rd}$ quartile range) unless indicated otherwise *indicating mean (SD).

SCIM, spinal cord injury independence measure; BMI, body mass index; BCTQ, boston carpal tunnel syndrome questionnaire.

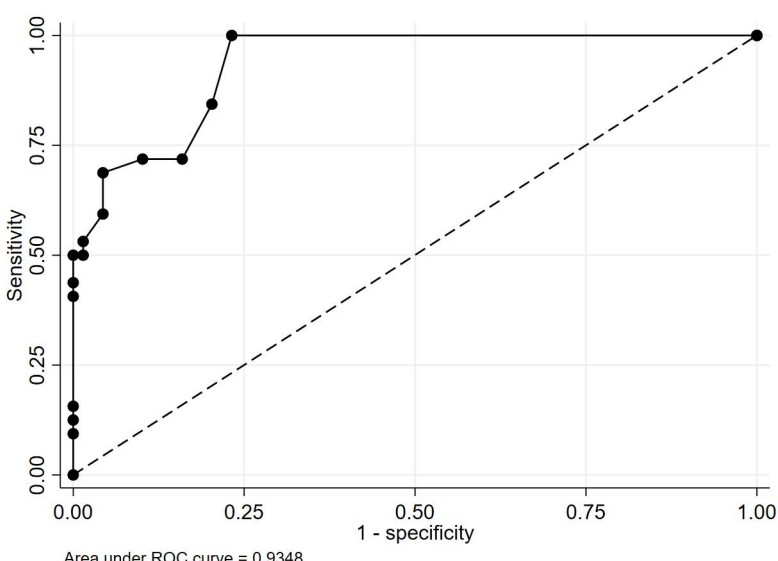

Area under ROC curve = 0.9348

**Fig 2. Area under the receiver operating characteristics curve (AuROC) of the CTS-6 Evaluation Tool for diagnosing CTS.**

performance remained the same for all three cutoff scores, whereas specificity performance differed minimally at 7.5 and 10 cutoff scores. Discriminative ability performance remained good, as demonstrated by AuROC, ranging from 0.719 (95%CI:0.631-0.806) to 0.750 (95%CI: 0.662-0.838) (Table 4).

## Discussion

This study was a geographic and domain validation of the CTS-6. Initially, the CTS-6 Evaluation Tool was developed in the United States from non-disabled participants, but this study

**Table 3. Diagnostic performance of the CTS-6 Evaluation Tool at three different cutoff points at 7.5, 10 and 12 (total n = 101).**

| CTS-6 cut off score | Sensitivity (95% CI) | Specificity (95% CI) | PPV (95% CI) | NPV (95% CI) | LR (95% CI) | AuROC (95% CI) |
|---|---|---|---|---|---|---|
| 7.5 (n = 25) | 68.8 (50.0-83.9) | 95.7 (87.8-99.1) | 88.0 (68.8-97.5) | 86.8 (77.1-93.5) | 15.8 (5.1-49.0) | 0.822 (0.737-0.907) |
| 10 (n = 16) | 50.0 (31.9-68.1) | 100.0 (94.8-100.0) | 100.0 (79.4-100.0) | 81.2 (71.2-88.8) | . | 0.750 (0.662-8.838) |
| 12 (n = 14) | 43.8 (26.4-62.3) | 100.0 (94.8-100.0) | 100.0 (76.8-100.0) | 85.1 (75.8-91.8) | . | 0.719 (0.631-0.806) |

PPV, positive predictive value; NPV, negative predictive value; LR, likelihood ratio; AuROC, area under the receiver operating characteristic curve.

**Table 4. Diagnostic performance of the CTS-6 Evaluation Tool at three different cutoff points: 7.5, 10, and 12 in the symptomatic subgroup (total n = 37).**

| CTS-6 cutoff score | Sensitivity (95% CI) | Specificity (95% CI) | PPV (95% CI) | NPV (95% CI) | LR (95% CI) | AuROC (95% CI) |
|---|---|---|---|---|---|---|
| 7.5 (n = 22) | 68.8 (50.0-83.9) | 80.0 (28.4-99.5) | 95.7 (78.1-99.9) | 28.6 (8.4-58.1) | 3.44 (0.586-20.2) | 0.744 (0.531-0.956) |
| 10 (n = 16) | 50.0 (31.9-68.1) | 100.0 (47.8-100.0) | 100.0 (79.4-100.0) | 23.8 (8.2-47.2) | ' | 0.750 (0.662-0.838) |
| 12 (n = 14) | 43.8 (26.4-62.3) | 100.0 (47.8-100.0) | 100 (76.8-100.0) | 21.7 (7.5-43.7) | . | 0.719 (0.631-0.806) |

PPV, positive predictive value; NPV, negative predictive value; LR, likelihood ratio; AuROC, area under the receiver operating characteristic curve.

applied this tool to wheelchair users in Northern Thailand. The diagnostic performance at 12 and 10 cutoff scores showed the greatest specificity of 100%; however, with limited sensitivity, unlike the lower cut-off score of 7.5, which showed moderate sensitivity, the specificity slightly decreased to 95.7%.

Because of the different population domain, this study's higher prevalence of CTS compared to non-disabled individuals potentially increased the PPV result. Nevertheless, this did not affect the sensitivity and specificity. As demonstrated by an AuROC of 0.9348, the exceptionally high discriminating ability might be due to incorporation bias, as the diagnostic reference standard for CTS in practice usually incorporates clinical symptoms plus supportive electrodiagnostic findings. The highest sensitivity result (68.8%) from a cut-off point of ≥ 7.5 would suit the screening role of the tool. With an increasing cut-off point, the specificity increased and reached 100% at a cut-off point of 10 and 12; with these cut-off points, the tool could help rule in CTS confidently. As an electrodiagnosis study usually requires another visit, clinicians with various expertise levels may feel more confident in reaching treatment decisions by applying a cutoff point of 10.

However, compared to the previous study by Fowler et al. [16], the sensitivity at a cutoff point of 12 was higher (95%), possibly due to different analytic methods, alternating clinical diagnosis by hand experts, electrodiagnosis, and ultrasound as the reference standard. A study by Makanji et al. [17] reported similar sensitivity (60%) and specificity (90%) in this study, using an electrodiagnostic study as the reference standard. This differed from the result of a study by Wang et al. [18] which used clinical diagnosis by hand expert as a reference standard; the reported sensitivity was 75% and specificity was 59%.

The study was limited in that the clinicians were rehabilitation specialists, not general practitioners or medical student trainees, which might have affected the tool's high discriminative ability and specificity. The performance of the tool might differ if the clinician has varying expertise. Another major limitation is the incorporation bias of the reference diagnostic

standard because CTS diagnosis requires clinical symptoms and confirmatory electrodiagnostic findings. The study reduced other potential biases, such as test review bias, by keeping the CTS-6 Evaluation Tool result from the rehabilitation physician performing electrodiagnosis, and vice versa. Partial verification bias was reduced, as all studied wrists underwent CTS-6 and electrodiagnostic tests. In addition, the diagnostic performance of the symptomatic subgroup remained similar to the analysis of total participants,

In conclusion, the CTS-6 Evaluation Tool demonstrated an excellent discriminative ability for CTS diagnosis in Thai wheelchair users population. Its role can vary when different cutoff points are applied. The original cutoff score of 12 did not perform as well as in previously developed studies in the non-disabled population domain. A cutoff score of 10 outperformed 12 for higher sensitivity with the same specificity. Therefore, it could be proposed as the cutoff score to rule in CTS for the wheelchair user population. A cutoff of 7.5 could also be suggested with a different application purpose; this cutoff is more suitable as a screening test for ruling out, as it gave moderately higher sensitivity but minimally less specificity, As the CTS-6 Evaluation Tool is a structured clinical interview and exam without the need for sophisticated investigation, it could be included as one of the long-term evaluation instruments for wheelchair users with long-term disability to help increase clinician awareness and early detection of potential underlying CTS. Further studies of the CTS-6 Evaluation Tool in the Thai population would benefit clinicians for more generalized applications. Potential for future studies include validation studies with Thai non-disabled individuals and reliability studies investigating different levels of clinician expertise.

## Supporting Information

**S1 Dataset. Dataset file underlying the findings described in the manuscript.**
(XLSX)

## Acknowledgements

The authors would also like to acknowledge this support.

-Rehabilitation Ward, Maharaj Nakorn Chiang Mai Hospital, Department of Rehabilitation Medicine, Faculty of Medicine, Chiang Mai University, for administrative support.

-The authors would like to thank all personnel of the Rehabilitation Ward, Maharaj Nakorn Chiang Mai Hospital Department of Rehabilitation Medicine, Center of Clinical Epidemiology and Clinical Statistics, Faculty of Medicine, Chiang Mai University, and all study participants.

-This study was partially supported by Chiang Mai University and the Faculty of Medicine, Chiang Mai University.

## Author contributions

**Conceptualization:** Montana Buntragulpoontawee, Siam Tongprasert, Jiraporn Khorana, Phichayut Phinyo.

**Data curation:** Montana Buntragulpoontawee, Jiraporn Khorana, Kittipong Kitisak, Waris Karinuntakul, Sineenard Pornjaksawan, Phichayut Phinyo.

**Formal analysis:** Montana Buntragulpoontawee, Jiraporn Khorana, Kittipong Kitisak, Phichayut Phinyo.

**Funding acquisition:** Montana Buntragulpoontawee.

**Investigation:** Montana Buntragulpoontawee, Siam Tongprasert, Kittipong Kitisak, Waris Karinuntakul, Sineenard Pornjaksawan.

**Methodology:** Montana Buntragulpoontawee, Siam Tongprasert, Jiraporn Khorana, Kittipong Kitisak, Waris Karinuntakul, Sineenard Pornjaksawan, Phichayut Phinyo.

**Project administration:** Montana Buntragulpoontawee, Kittipong Kitisak, Waris Karinuntakul, Sineenard Pornjaksawan.

**Software:** Jiraporn Khorana, Kittipong Kitisak, Sineenard Pornjaksawan, Phichayut Phinyo.

**Supervision:** Siam Tongprasert, Jiraporn Khorana, Phichayut Phinyo.

**Validation:** Kittipong Kitisak, Phichayut Phinyo.

**Visualization:** Phichayut Phinyo.

**Writing – original draft:** Montana Buntragulpoontawee, Phichayut Phinyo.

**Writing – review & editing:** Montana Buntragulpoontawee, Siam Tongprasert, Jiraporn Khorana, Kittipong Kitisak, Waris Karinuntakul, Sineenard Pornjaksawan, Phichayut Phinyo.

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
