## [Decision Letter · Decision Letter 0]

26 Nov 2024

PONE-D-24-40688The validation of the CTS-6 evaluation tool for diagnosing carpal tunnel syndrome (CTS) in Thai wheelchair usersPLOS ONE

Dear Dr. Phinyo,

Thank you for submitting your manuscript to PLOS ONE. After careful consideration, we feel that it has merit but does not fully meet PLOS ONE’s publication criteria as it currently stands. Therefore, we invite you to submit a revised version of the manuscript that addresses the points raised during the review process.

We look forward to receiving your revised manuscript.

Kind regards,

Paraskevopoulos Eleftherios

Academic Editor

PLOS ONE

Journal Requirements:

2. Thank you for stating the following financial disclosure: The Faculty of Medicine at Chiang Mai University provided a grant for The Entrapment Neuropathy in Thai Wheelchair Users due to SCI Registry. registry (grant number 046/2565).  

3. In the online submission form, you indicated that to comply with the institution's policy, the data underlying the results presented in the study are available upon request.

Reviewers' comments:

Reviewer's Responses to Questions

**Comments to the Author**

1. Is the manuscript technically sound, and do the data support the conclusions?

Reviewer #1: Yes

Reviewer #2: Yes

2. Has the statistical analysis been performed appropriately and rigorously? 

Reviewer #1: Yes

Reviewer #2: Yes

3. Have the authors made all data underlying the findings in their manuscript fully available?

Reviewer #1: Yes

Reviewer #2: Yes

4. Is the manuscript presented in an intelligible fashion and written in standard English?

Reviewer #1: Yes

Reviewer #2: Yes

5. Review Comments to the Author

Reviewer #1: Phinyo et al aims to validate the CTS-06 tool, which has been previously validated in in disabled patient, in a population of patients with disability. This adds to the literature regarding the usefulness of the CTS-6 in a no general population class. They show good discriminative ability of CTS-6 in this different population. Overall, interesting study and paper with some minor issues.

1.There appears to be a formatting issue in Table 2 and 3 with the parentheses appearing backwards throughout the table.

2.The conclusion should specify that this study was performed in a Thai or Asian population, as findings of these results may not possibly apply to other countries where there is notable difference in population (heterogeneity in population, higher overall BMI, higher use of electric wheelchairs, higher comorbidities (DM for which it appears there were none in this study unlike in more westernized countries, ...)

3.They report that the BMI appear to be higher but is not statistically significant, ergo meaning without context would indicate there were no difference. They should report the latter or add the context that CTS is known to be associated with higher BMI (with at least one citation)

4.Comparison with normal Thai population would have added to the findings and interpretation of the data in order to better compare difference between non disabled and disabled patient. but I understand this may not be possible, overly cumbersome.

Reviewer #2: Overall, research question and methodology sound good.

I had minor comment in conclusion part. I agreed with the authors that “A cutoff of 7.5 could also be suggested with a different application purpose; this cutoff is more suitable as a screening test for ruling out, as it gave moderately higher sensitivity but minimally less specificity, As the CTS-6 evaluation tool is a structured clinical interview and exam without the need for sophisticated investigation”. This point also should be summarized in the conclusion of the abstract part.

There was minor error in Table 2 and 3 in Typing with ( ) such as Age )year(* , )person(.

6. PLOS authors have the option to publish the peer review history of their article (what does this mean? ). If published, this will include your full peer review and any attached files.

**Do you want your identity to be public for this peer review?** For information about this choice, including consent withdrawal, please see our Privacy Policy .

Reviewer #1: No

Reviewer #2: No

---

## [Author Response · Author response to Decision Letter 1]

6 Jan 2025

Dear Editor,

The authors would like to thank the editor and reviewers (Reviewers#1 and #2) for reviewing and providing valuable suggestions to the manuscript entitled “The Validation of the CTS-6 Tool for Diagnosing Carpal Tunnel Syndrome (CTS) in Thai Wheelchair Users” submitted to PLOS ONE on 13th September 2024. We have addressed their comments and responded accordingly. The indicated location of the revised content refers to the “Revised Manuscript with Track Changes” file. Please find the point-by-point responses below.

Thank you.

Reviewer reports:

Reviewer #1:

1. Phinyo et al aims to validate the CTS-06 tool, which has been previously validated in in disabled patient, in a population of patients with disability. This adds to the literature regarding the usefulness of the CTS-6 in a non-general population class. They show good discriminative ability of CTS-6 in this different population. Overall, interesting study and paper with some minor issues.

Response: The authors thank the reviewer for positive comments and suggestions to improve the manuscript.

2. There appears to be a formatting issue in Table 2 and 3 with the parentheses appearing backwards throughout the table.

Response: We found that this was a technical formatting issue with Microsoft Word. Now we have it resolved.

3. The conclusion should specify that this study was performed in a Thai or Asian population, as findings of these results may not possibly apply to other countries where there is a notable difference in population (heterogeneity in population, higher overall BMI, higher use of electric wheelchairs, higher comorbidities (DM for which it appears there were none in this study unlike in more westernized countries, ...)

Response: We added the specific population in which this study performed on in the conclusion part, page 15, paragraph 2, line 294

4. They report that the BMI appear to be higher but is not statistically significant, ergo meaning without context would indicate there were no difference. They should report the latter or add the context that CTS is known to be associated with higher BMI (with at least one citation)

Response: We added the context about CTS and its association with higher BMI at page 10 paragraph 1 line 206 with citation at line 209.

5. Comparison with normal Thai population would have added to the findings and interpretation of the data in order to better compare difference between non disabled and disabled patient. but I understand this may not be possible, overly cumbersome.

Response: Thank you for your insightful suggestion. We would take this point to considering when planning our future study.

Reviewer #2:

1. Overall, research question and methodology sound good. I had minor comment in conclusion part. I agreed with the authors that “A cutoff of 7.5 could also be suggested with a different application purpose; this cutoff is more suitable as a screening test for ruling out, as it gave moderately higher sensitivity but minimally less specificity, As the CTS-6 evaluation tool is a structured clinical interview and exam without the need for sophisticated investigation”. This point also should be summarized in the conclusion of the abstract part.

Response: We have summarized the suggested point in the conclusion of the abstract part at page 2 paragraph 4 line 43-44 and 47-48.

2. There was minor error in Table 2 and 3 in Typing with ( ) such as Age )year(* , )person(.

Response: We found that this was a technical formatting issue with Microsoft Word. Now we have it resolved.

---

## [Editor Report · Decision Letter 1]

29 Jan 2025

The validation of the CTS-6 evaluation tool for diagnosing carpal tunnel syndrome (CTS) in Thai wheelchair users

PONE-D-24-40688R1

Dear Dr. Phichayut Phinyo,

We’re pleased to inform you that your manuscript has been judged scientifically suitable for publication and will be formally accepted for publication once it meets all outstanding technical requirements.

Kind regards,

Paraskevopoulos Eleftherios

Academic Editor

PLOS ONE
---

## [Editor Report · Acceptance letter]

PONE-D-24-40688R1

PLOS ONE

Dear Dr. Phinyo,

I'm pleased to inform you that your manuscript has been deemed suitable for publication in PLOS ONE. Congratulations! Your manuscript is now being handed over to our production team.

Kind regards,

on behalf of

Dr. Paraskevopoulos Eleftherios

Academic Editor

PLOS ONE